# Microplanning improves stakeholders' perceived capacity and engagement to implement lymphatic filariasis mass drug administration

**Caitlin M. Worrell**[1,2,3]*, **Tara A. Brant**[1], **Alain Javel**[4], **Eurica Denis**[4], **Carl Fayette**[4], **Franck Monestime**[4], **Ellen Knowles**[5], **Cudjoe Bennett**[5], **Jürg Utzinger**[2,3], **Peter Odermatt**[2,3], **Jean-Frantz Lemoine**[6]

**1** United States of America Centers for Disease Control and Prevention, Atlanta, Georgia, United States of America, **2** Swiss Tropical and Public Health Institute, Allschwil, Switzerland, **3** University of Basel, Basel, Switzerland, **4** IMA World Health, Port-au-Prince, Haiti, **5** IMA World Health, Washington, District of Columbia, United States of America, **6** National Program to Eliminate Lymphatic Filariasis, Ministry of Public Health and Population, Port-au-Prince, Haiti

* uvz2@cdc.gov

## Abstract

### Background

Achieving adequate mass drug administration (MDA) coverage for lymphatic filariasis is challenging. We sought to improve stakeholder engagement in MDA planning and increase subsequent MDA coverage through a series of microplanning workshops.

### Methodology

Prior to the 2018 MDA, Haiti's Ministry of Public Health and Population (MSPP) and partners conducted 10 stakeholder microplanning workshops in metropolitan Port-au-Prince. The objectives of the workshops were to identify and address gaps in geographic coverage of supervision areas (SAs); review past MDA performance and propose strategies to improve access to MDA; and review roles and responsibilities of MDA personnel through increased stakeholder engagement. Retrospective pre-testing was employed to assess the effectiveness of the workshops. Participants used a 5-point scale to rank their understanding of past performance, SA boundaries, roles and responsibilities, and their perceived engagement by MSPP. Participants simultaneously ranked their previous year's attitudes and their attitudes about MDA following the 2-day microplanning workshop. Changes in pre- and post-scores were analyzed using Wilcoxon signed-rank tests.

### Principal findings

A total of 356 stakeholders across five communes participated in the workshops. Participants conducted various planning activities, including revising SA boundaries to ensure full geographic reach of MDA, proposing or validating social mobilization strategies, and proposing other MDA improvements. Compared with previous year rankings, the

**Data availability statement:** The authors confirm that the data supporting the findings of this study are available within the article and its supplementary materials.

**Funding:** The project was supported with U.S. Department of Health and Human Services appropriated funding through a co-operative agreement to IMA World Health through Award #U51GH000952 (AJ, EU, DF, FM, EK, CB). CW and TB are employees of the U.S. Centers for Disease Control and Prevention. The findings and conclusions in this report are those of the author(s) and do not necessarily represent the official position of the Centers for Disease Control and Prevention. The funders had no role in study design, data collection and analysis, decision to publish, or preparation of the manuscript.

**Competing interests:** The authors have declared that no competing interests exist.

workshops increased participant understanding of past performance by 1.34 points (standard deviation [SD] = 1.05, p <0.001); SA boundaries by 1.14 points (SD = 1.30; p <0.001); their roles and responsibilities by 0.71 points (SD = 0.95, p <0.001); and sense of engagement by 1.03 points (SD = 1.08, p <0.001). Additionally, compared with 2017, drug coverage increased in all five communes during the 2018 MDA.

## Conclusions/significance

Participatory stakeholder workshops during MDA planning can increase self-reported engagement of key personnel and may improve staff performance and contribute to achievement of drug coverage targets. Microplanning success was supported by MDA results, with all communes achieving preset MDA coverage targets.

## Author summary

Lymphatic filariasis is a neglected tropical disease that can be eliminated by periodically treating entire at-risk communities with safe and efficacious medicines, a strategy known as mass drug administration (MDA). Prior experience shows that MDA campaigns require intense planning to ensure that every eligible person within the community can receive the medicines. We aimed to improve the campaign by better involving key MDA stakeholders such as volunteers and other important community members in the planning process, through microplanning. The participants suggested many strategies to help the campaign reach more community members, including better ways to prepare and inform the community that the campaign is happening. We invited the microplanning participants to provide feedback on how this new strategy worked compared with their experiences during past campaigns. Participants reported that they felt better engaged by health authorities, and in particular, that they had more information about the results of past campaigns, where they should be distributing medicines, and their specific tasks and responsibilities during the campaign. We found that more people received medicines during the campaign that followed the microplanning workshops compared with the previous campaigns. We conclude that microplanning helped to increase the number of people who received MDA medicines.

## Introduction

Lymphatic filariasis (LF) programs in some countries have met global program targets for elimination. However, many programs continue to face challenges in achieving high levels of community participation in mass drug administration (MDA) [1]. Triple drug regimen MDA [2], with diethylcarbamazine (DEC), ivermectin (IVM), and albendazole (ALB), has been shown to accelerate national and global progress toward LF elimination in several contexts [3]. Based on this success, Haiti has adopted triple drug therapy as the preferred treatment strategy in Haiti, however, the benefits of such novel approaches can only be realized if a high proportion of at-risk community members have access to the intervention and are receptive to taking the drugs [3]. Numerous factors can impact community drug coverage and compliance [4,5], and the factors that influence an individual's willingness and ability to participate in MDA are multi-factorial and subject to change through multiple rounds of MDA [5]. These factors are often related to the broader social-ecological context, as well as provider or recipient community characteristics.

While many studies have highlighted the recipient community characteristics that drive non-adherence to MDA [6–8], increasing focus has been placed on understanding and addressing program-related issues that are important drivers of drug uptake or lack thereof [9]. In Tanzania, for example, the majority of community members who did not receive medication reported program-related issues, such as community drug distributors (CDDs) not visiting all eligible households, inopportune timing of the MDA which meant that many people were at their farms at the time of the MDA, and not knowing that MDA was occurring [10]. Other reported programmatic issues associated with poor MDA compliance include drug shortages, insufficient time and human resources to achieve coverage targets, inadequate training of drug distributors, inappropriate selection of drug distributors, poor adverse event management, and limited community engagement or coordination in MDA planning [5]. Failure to adequately engage members of the health system and the community during MDA planning can adversely impact the quality of MDA implementation, and hence, lead to low levels of community participation [11,12]. Conversely, strong community participation in the planning and implementation of MDAs has been seen as essential to the success of some campaigns, particularly in urban areas [13–15]. In India, the involvement of mid-level health authorities, especially at the district level, was identified as a key element in carrying out a successful MDA [6].

Microplanning is an approach that may improve access to, and acceptability of, MDA programs by addressing the various issues faced by providers and recipients [16]. While the specific implementation varies by location, microplanning generally encompasses a 'bottom-up' approach that engages local stakeholders in health program planning by leveraging local data and knowledge to both identify site-specific problems and design solutions [17]. This iterative and multi-stage process engages local representatives to define the target populations in a delineated area that are eligible for an intervention [18]. During this process, stakeholders define the location of the target population and the catchment areas for each program delivery actor. Stakeholders design service delivery strategies to reach targeted sub-populations, often with an emphasis on hard-to-reach populations (e.g., mobile and remote populations) [19,20]. They create a realistic operational plan based on local circumstance and available resources, as well as determine the material, financial, and human resources needed to accomplish program targets. Microplans are typically gathered and compiled across various administrative levels and, ideally, inform the allocation of program resources.

Microplanning has been shown to improve public health program coverage [21–24], and is a key component of the Reaching Every District strategy [18,25] for improving immunization systems and vaccination coverage [21,26,27]. Outside immunization programs, microplanning has been used in planning and delivering other health interventions, including HIV care and prevention [28,29], malaria prevention [23,30,31], and reproductive and child health programs [32,33]. Neglected tropical disease (NTD) programs, such as LF MDA, have applied microplanning less frequently than other programs. However, the World Health Organization (WHO) recommends microplanning for NTD programs conducting MDA, with an emphasis on areas where MDA coverage has been consistently low, where there is evidence of populations who have never been treated, or where assessments show that LF transmission persists despite multiple rounds of treatment [16].

Digital tools and geospatial mapping are increasingly seen as a critical component to microplanning, especially as a tool that can accurately and efficiently guide programs in defining the location and characteristics of communities for the purposes of health program planning and implementation [30,34]. These tools have been shown to be a cost-effective planning strategy [30,35] that can increase program coverage [23,28,36], identify mobile and displaced populations [26,34,36], and promote an efficient allocation of resources [37]. Including these strategies in microplanning efforts in complex and highly dynamic settings, such as Port-au-Prince, were seen as critical for ensuring efficient and representative microplanning.

Haiti has been implementing large scale MDA campaigns for LF since 2000 and reached full scale in 2012 once MDA began in the metropolitan region of Port-au-Prince [38]. Despite most of the communes in Port-au-Prince achieving the globally recommended target of at least 65% population coverage of MDA for LF during the first MDA round [39], the Port-au-Prince communes struggled to achieve coverage targets during subsequent rounds of MDA [40]. In 2016, sentinel site assessments identified that five of six Port-au-Prince communes required additional rounds of MDA that achieved coverage targets prior to qualifying to undergo the transmission assessment survey (TAS). After the 2017 MDA, in which coverage targets were again not met, the Ministry of Public Health and Population (MSPP) convened a meeting with implementing partners to create a plan for improving coverage in subsequent MDAs. During these meetings, MSPP and partners identified a series of strategies for improving coverage based on literature of best MDA practices in urban areas and lessons learned by MSPP staff and partners during years of MDA implementation. Microplanning was one of several proposed activities.

The project reported here details the results of a microplanning activity aimed to improve 2018 MDA coverage in the five communes of Port-au-Prince that had experienced declining MDA coverage. Our specific objectives were to evaluate the effect of using established microplanning techniques on (i) identifying program gaps and key solutions; (ii) the perceived performance and engagement of key stakeholders (both community leaders and program volunteers) who attended a microplanning workshop; and (iii) MDA coverage in the subsequent 2018 MDAs conducted in five communes of Port-au-Prince.

## Methods

### Ethics statement

This project took place as an evaluation of a routine LF elimination program and was considered by Haiti's MSPP to be a program evaluation. The project is covered by a non-research determination granted by the Center for Global Health (CGH) Human Subjects Protection Office at the U.S. Centers for Disease Control and Prevention (CDC) in Atlanta, Georgia, United States of America (project identification #0900f3eb81b9b8d5). In line with the non-research determination, written consent was not obtained since this project was determined to be a routine public health evaluation activity.

### Project area

This evaluation took place from January through March 2018 in five of the six communes of metropolitan Port-au-Prince (i.e., Carrefour, Cité-Soleil, Delmas, Port-au-Prince, and Tabarre) that required MDA as of 2018. This area, which represents a population of approximately 2.3 million people, was the last area in Haiti to initiate LF MDA in 2012 [38].

In Port-au-Prince, MDA is delivered through distribution posts located in the community and at schools. One post, staffed by three CDDs, is charged with treating approximately 1,000 people over a 4-day MDA. Community promoters (CPs) manage approximately three health posts and are in turn managed by community leaders (CLs). CLs, with assistance from CPs, are responsible for conducting social mobilization activities within a supervision area (SA) in the months prior to the MDA, selecting and supervising distribution sites, and training CPs and CDDs.

### Project design

To improve stakeholder engagement and MDA implementation, the microplanning team, comprised of MSPP leadership and implementing partner staff with technical assistance from staff from CDC, conducted a 2-phase microplanning exercise in the five target communes. In phase one, the microplanning team in collaboration with CLs, conducted an inventory of

the 2017 MDA distribution posts using Open Data Kit (ODK) collect application (Open Data Kit Inc., v2.0) [41] loaded onto mobile devices. Historic distribution post sites were overlaid onto a Google satellite base layer (Google Inc, n.d.), using open-source QGIS (Open Source Geospatial Foundation Project, v2.14.21). To identify possible access gaps, we applied a 300 m buffer around each distribution post to identify areas within approximately a 500 m maximum walking distance (S1 Fig). These maps were prepared to inform the discussion during microplanning workshops in phase two.

During phase two, all CLs as well as a selection of CPs, and other community stakeholders were invited to participate in microplanning workshops. In close consultation with communal and departmental focal points from MSPP, the microplanning team identified key community stakeholders who are responsible for supporting the distribution of MDA in schools or in the broader community. These individuals included school directors, and Ministry of National Education and Vocational Training inspectors at the district and sub-district level. For community-based MDA this included representatives from local authorities such as mayors and local council members (*Conseil d'Administration de la Section Communale* – CASECs), church-leaders and pastors, voodoo temple priests or *Ougans*, and local youth associations. The workshop was moderated by communal and departmental MSPP staff with support from implementing partner staff who recorded and synthesized the discussion and suggested action points. The microplanning workshops included several key activities. Initially, the microplanning team provided an overview of the LF elimination program objectives, reviewed the historical results from the 2012 to 2017 MDAs, summarized the results of several key MDA evaluations, and engaged the participants in a discussion of the challenges to, and opportunities for, improving MDA coverage. To clarify stakeholder roles and responsibilities, the microplanning team presented and validated the terms of references for key MDA volunteers (i.e., CLs, CPs, and CDDs), and drafted a document detailing expectations from opinion leaders and other stakeholders. Further, microplanning team members presented several specific strategies that were proposed by implementing partners to improve the 2018 MDA. These strategies included but were not limited to extending the number of MDA days, modifying MDA distribution times, enhancing schools' participation in MDA, improving visibility and credibility of CDDs, and expanding mop-up activities. Participants were invited to propose revisions including defining and validating key strategies for CDDs (e.g., optimal timing and location).

Finally, SA boundaries were delineated for each CL in QGIS. Global positioning system (GPS) coordinates of distribution posts collected during phase one were overlayed onto an Open Street Maps layer that included various landmarks (e.g., streets, places of business, etc.). Google Maps and satellite base layers (Google Inc., n.d.) were used as additional references for identifying landmarks and delimiting boundaries. Through an iterative process, microplanning participants used the 2017 distribution post maps as a basis to discuss and agree upon the SA boundaries for each CL, ensuring no overlapping or omission of areas within the targeted communes.

## Impact measures

We used retrospective pre-testing to evaluate the microplanning workshops against the stated objectives [42–44]. This method involves asking participants in a single data collection event to rate their attitudes before and after the workshop. Following the microplanning workshops, participants were invited to complete a questionnaire simultaneously rating their previous year's perceptions and their perceptions following the 2-day microplanning workshop. This strategy aimed to reduce the confounding factor of response shift bias [45], a phenomenon where the respondents' internal frame of reference is altered due to the influence of the

intervention itself. In this context, the participants perceptions of their previous knowledge and engagement may be altered by participating in the workshop, as has been seen in evaluations using traditional self-reported pre-post-test evaluation frameworks [46]. We chose four metrics that we believe were important for participants success and their perceived engagement, which we saw as a key motivator to participate in the MDA and achieve high coverage. Participants were asked to evaluate four measures: (i) understanding past performance; (ii) understanding the boundaries of coverage zones or area of influence; (iii) understanding roles and responsibilities in the MDA; and (iv) perceived level of engagement by MSPP in the MDA planning process. Participant commune and role in the MDA were also recorded. Each measure was evaluated using a 5-point scale (i.e., 1 = poor; 2 = fair; 3 = good; 4 = very good; and 5 = excellent; N/A = not applicable). We also assessed changes in coverage between the 2017 and 2018 MDAs as an indirect measure of impact. MDA coverage was calculated using the total number of MDA doses delivered by each commune's MDA distribution teams divided by the estimated population of the commune.

## Statistical analysis

Data were compiled using spreadsheet software (Microsoft Excel 2010, Microsoft; Seattle, Washington, United States of America) and data management and statistical analyses were completed using SAS v9.4 (SAS Institute; Cary, North Carolina, United States of America). For pre-workshop and post-workshop summary analyses, all responses were considered; however, results were filtered to include only complete pre-post pairs to measure changes as a result of the workshop. Within pair differences between the pre- and post-workshop ratings were analyzed using a Wilcoxon signed-rank test at the 95% confidence level to determine whether microplanning had impacted perceived capacity to deliver quality MDA as well as key engagement metrics. Alluvial plots describing changes in participant ratings were created in R v1.1.463 (RStudio, Free Software Foundation Inc.; Boston, Massachusetts, United States of America). Maps were created using ArcGIS software (ESRI v10.6; Redlands, California, United States of America) and QGIS (Open Source Geospatial Foundation Project, v.3.38.3).

## Results

### Microplanning outputs

During phase one, each of the 73 CLs was accompanied by a GPS data enroller who captured 8,034 GPS coordinates collectively at the locations of each of the 2017 distribution points. In advance of the microplanning workshop, the GPS coordinates were plotted in QGIS overlayed on satellite images of Port-au-Prince to create a detailed distribution post map for all five communes. Microplanning team members used these distribution post inventory maps to assess coverage gaps that were presented to MDA stakeholders.

During phase two, a total of ten 2-day microplanning workshops were held for 356 participants, including 73 CLs, 240 CPs, and 43 other key MDA stakeholders selected from the five targeted communes. Several activities were conducted as part of the workshops. First, the microplanning team presented workshop participants with the maps showing the location of the 2017 distribution posts. They then worked with CLs to delineate the 2017 SA boundaries as they were understood by each CL. S2 Fig shows the resulting SA maps following the iterative process by which the microplanning team and workshop participants modified the MDA maps to ensure no overlapping or omitted areas within the targeted communes. As part of this process, certain leaders' distribution posts that fell outside their updated SA boundaries were flagged to be prioritized for relocation.

To support the optimal deployment of posts, each CL received a printed map following the workshop that contained detailed information about their SA, including roads and points of interest such as churches, businesses, and schools (Fig 1). Each CL's SA was clearly outlined, and neighboring CLs were listed. Maps were printed in Haitian Creole to improve the ease of use by stakeholders and listed the CL's name to foster a sense of ownership over the activities in that area. Fig 1 shows an illustrative CL's supervision map.

Another key activity included reviewing past MDA performance, identifying local challenges to achieving high coverage, and proposing updated strategies to improve program coverage. As part of this process, participants discussed and finalized distribution strategy modifications including some modifications proposed by the microplanning team, as well as drafts of new social mobilization strategies and tools that were developed by LF program staff and partners. Table 1 describes a selection of key discussion points and associated solutions based on these discussions. Participants identified key actions to improve partnerships, planning, and logistics associated with MDA planning; strategies to improve support for CLs, CPs,

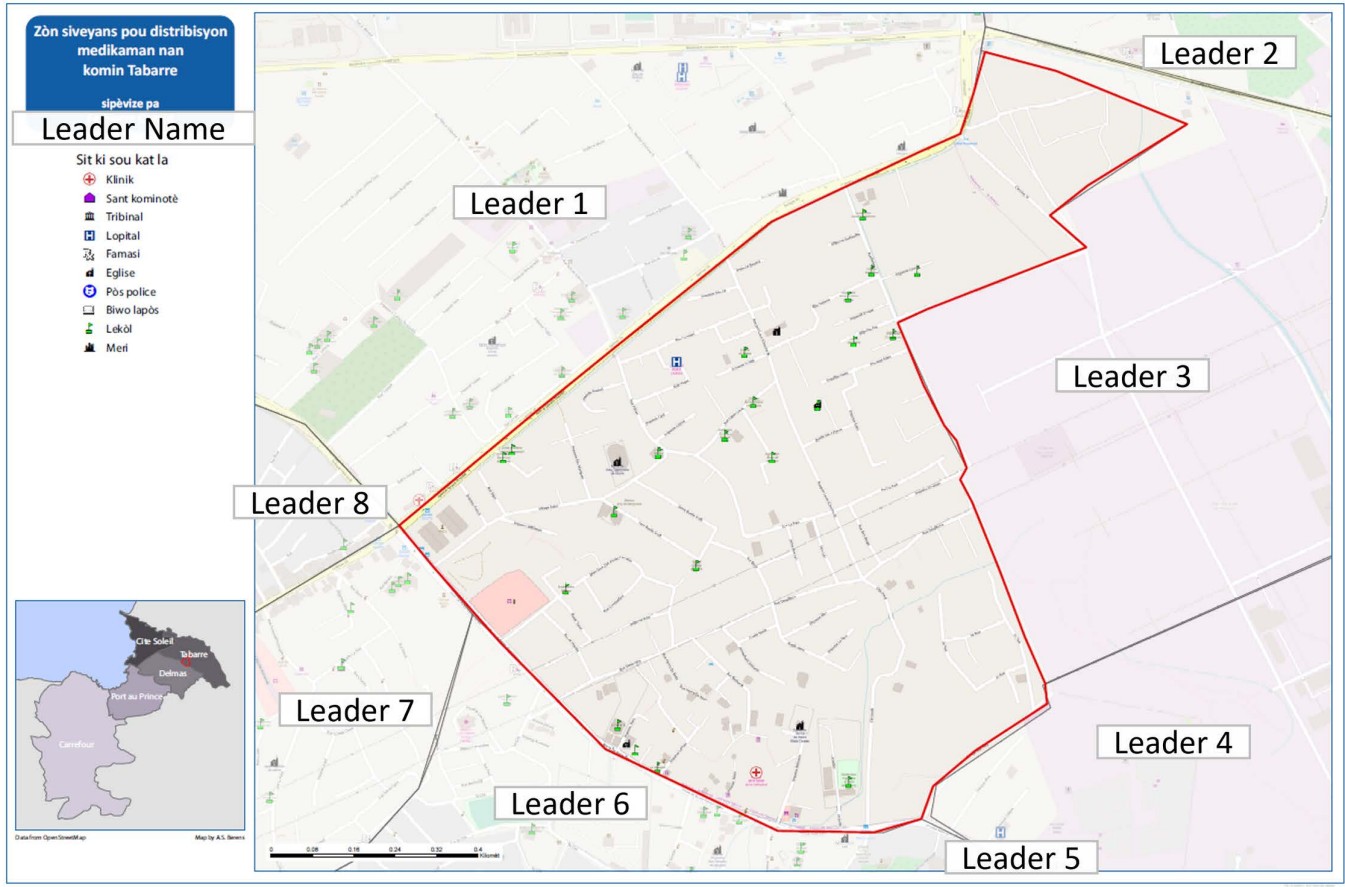

**Fig 1. Illustrative supervision area (SA) maps, 2018 mass drug administration (MDA) in Port-au-Prince, Haiti.** Illustrative SA map displaying the SA for a community leader (CL) in the commune of Tabarre for the 2018 MDA for lymphatic filariasis (LF) in Haitian Creole. The map includes the boundaries of the SA (in red) including infrastructure and key points of interest. Additionally, it shows the CLs working in adjacent SAs. For privacy reasons, this figure has been lightly edited to anonymize the names of CLs. This figure uses a basemap obtained from OpenStreetMaps. The base map is available for download at: https://nominatim.openstreetmap.org/ui/details.html?osmtype=R&osmid=307829&class=boundary under the following terms and conditions: https://osmfoundation.org/wiki/Terms_of_Use. The map of Fig 1 was created using ArcGIS software by Esri (www.esri.com). ArcGIS and ArcMap are the intellectual property of Esri and are used herein under license. Copyright Esri. All rights reserved. For more information about Esri software, please visit www.esri.com.

**Table 1. Selected feedback and outcomes from microplanning workshops conducted in Port-au Prince, Haiti in 2018.**

| Theme | Identified issues/ key suggestions | Decisions or actions taken[1] |
|---|---|---|
| *Partnerships, planning, and logistics* | Some school authorities refuse to participate in MDA activities due to lack of timely official correspondence from Ministry officials | Departmental authorities will prepare official correspondence detailing the specifics of the MDA distribution. This correspondence will be available to MDA staff within 1 week of the microplanning workshops |
| | Local authorities should accompany leaders during mobilization and promotion events at schools, churches, and other settings | Local authorities are involved in the microplanning workshops to ensure that they are informed of the MDA activities and may coordinate with MDA staff to mobilize and sensitize their populations |
| | Boundaries of SAs are poorly understood by leaders and other authorities | SA boundaries are defined during the microplanning workshops. Detailed maps of the newly defined SAs, including boundaries, infrastructures, and key points of interest, will be distributed to leaders to facilitate efficient MDA planning |
| *Supporting leaders, promoters, and CDDs* | Improving volunteer conditions and increasing the perceived legitimacy of the MDA volunteers by the community would help the LF program to achieve better results | CLs and CPs will be provided official badges to facilitate their identification and perceived legitimacy, particularly at institutions such as schools |
| | Existing social mobilization tools are outdated and insufficient for adequate social mobilization of the community | New social mobilization tools will be created as part of broader MDA strengthening activities. Training sessions will be held for the main MDA actors orienting them on the use of new social mobilization tools |
| *Social mobilization strategies* | The program should create testimonials or documentaries featuring persons affected by LF | Radio spots will be released promoting the MDA that include testimonials of persons affected by LF |
| | Local celebrities and CLs should be better engaged in social mobilization campaigns | A local comedian will be engaged to create social mobilization messages that will be disseminated on social media platforms[2] |
| | High level officials and medical professionals should take medications in public to improve public confidence in the reliability of the medicines | A public launch of the MDA will be held and local authorities and partners will swallow the MDA medicines in public |
| | The program should publicize the location of sites that can manage adverse events during the MDA | The program will make adverse event management information available through a toll-free hotline |
| *Distribution strategies* | Schools are a key population that should be emphasized during MDA; however, limited school information is available for planning (e.g., number of teams needed, quantity of medicines, and supplies), including lists and enrollment numbers of schools in the treatment zone | Schools will be prioritized during MDA. Data on school population will be collected in collaboration with the Ministry of National Education and Vocational Training (MENFP) |
| | MDA distribution hours should be shifted to include early evening hours to capture populations who are working or unavailable during working hours | Several communes will shift daily distribution times from 09:00–17:00 to 10:00–18:00 hours; however, other communes will maintain the original distribution hours due to security concerns |
| | Leaders and drug distributors have difficulty reaching populations who live/work in restricted access zones (e.g., industrial areas or gated communities) | For leaders affected in restricted access zones, a specific distribution strategy will be developed with the list of key partners in support |

*(Continued)*

**Table 1.** (Continued)

| Theme | Identified issues/ key suggestions | Decisions or actions taken[1] |
|---|---|---|
| | Some zones have insufficient personnel to adequately serve the targeted populations | Two additional CPs will be assigned to support MDA activities in the Grand Ravine and Tibwa area of Port-au-Prince commune |
| | Upon harmonization of SA boundaries, stakeholders recognized that several MDA distribution posts were located in neighboring SAs or communes | Several leaders' posts were identified for re-location. A Cité-Soleil leader will supervise posts in bordering Port-au-Prince, due to hesitation of Port-au-Prince leaders to work there due to insecurity. However, these posts will be counted as part of the Port-au-Prince coverage figures |
| | Several areas had insufficient distribution post coverage (e.g., mountainous areas within Carrefour commune) | Additional distribution posts will be added to fill identified coverage gaps. Further, distribution posts will be reorganized to provide better coverage of the SAs, particularly in the mountainous areas of Carrefour commune. This includes moving distribution posts closer to population centers and adding distribution areas in areas with insufficient coverage |
| | Mop-up should be conducted after the initial MDA campaign to maximize coverage | Both passive and active mop-up strategies will be implemented after the 5-day MDA campaign |

[1] Some actions described in this table were devised by partners based on feedback from the microplanning process or other MDA improvement activities in response to stakeholder feedback and suggestions

[2] Link to social media https://www.instagram.com/p/BifOhCVBfFM/?utm_source=ig_web_copy_link

and CDDs; new social mobilization strategies; as well as modifications to improve distribution strategies.

Several challenges were identified that were unable to be addressed fully by the LF program at the time of the activity for a variety of reasons, including financial constraints, conflicts with program policies, or feasibility of scaling the suggested intervention. For example, participants from multiple communes noted a need to improve working conditions for the leaders to achieve better results, noting a particular need for umbrellas to provide shade at distribution posts and increasing MDA stipends. Participants from Carrefour noted that lack of health facilities in rural sections of communes pose a challenge for adverse events management. Participants from Cité-Soleil noted the need to combat LF at different stages in the transmission cycle, especially as effluence from factories creates sanitation concerns within the commune. Finally, the microplanning team presented and validated the roles and responsibilities of CLs, CPs, and CDDs.

## Respondent evaluation

Following the workshop, participants simultaneously evaluated their perceptions of their capacity and engagement prior to, and following the, microplanning workshops using a retrospective pre-test model. Evaluation questionnaires were completed by 339 (95.2%) workshop participants across all five targeted communes, including two (0.6%) national-level MSPP staff, two (0.6%) departmental-level MSPP staff, 68 (20.1%) CLs, 227 (67.0%) CPs, and 40 (11.8%) community members (Table 2). Participant counts were approximately proportional to the underlying commune population.

At baseline, understanding roles and responsibilities had the highest score 3.91 ± 0.91 (mean ± standard deviation, SD), followed by perceived engagement 3.41 ± 1.07, SAs 3.29 ± 1.12, and past performance 3.05 ± 1.00 (Table 3). The highest proportion of individuals providing a "poor" assessment was seen within respondents' understanding of the boundaries of SAs (n = 8, 21%), while the highest proportion of individuals reporting "excellent" was seen with understanding roles and responsibilities (n = 73, 28%).

Table 2.  Characteristics of microplanning workshop participants in Port-au-Prince, Haiti in 2018 (n = 339).

| Characteristics | No. (%) of respondents |
|---|---|
| **Commune** | |
| Carrefour | 72 (21.2%) |
| Cité-Soleil | 70 (20.7%) |
| Delmas | 73 (21.5%) |
| Port-au-Prince | 96 (28.3%) |
| Tabarre | 28 (8.3%) |
| **Role** | |
| National MSPP | 2 (0.6%) |
| Departmental MSPP | 2 (0.6%) |
| Community leader (CL) | 68 (20.1%) |
| Community promoter (CP) | 227 (67.0%) |
| Community member | 40 (11.8%) |

Table 3.  Retrospective pre-testing results collected following microplanning workshops held in Port-au-Prince, Haiti in 2018.

| | Pre-workshop | | Post-workshop | | Difference (post-pre) | | |
|---|---|---|---|---|---|---|---|
| | n | Mean (SD) | n | Mean (SD) | n* | Mean (SD) | p-value |
| **Past performance** | 303 | 3.05 (1.00) | 302 | 4.37 (0.72) | 283 | 1.34 (1.05) | <0.001 |
| **Roles/responsibilities** | 277 | 3.91 (0.91) | 283 | 4.61 (0.60) | 262 | 0.71 (0.95) | <0.001 |
| **Supervision areas** | 277 | 3.29 (1.12) | 286 | 4.39 (0.82) | 266 | 1.14 (1.30) | <0.001 |
| **Perceived engagement** | 281 | 3.41 (1.07) | 286 | 4.45 (0.79) | 268 | 1.03 (1.08) | <0.001 |

*Only individuals with complete pre- and post-workshop data were considered for a difference in pre- and post-workshop scores. Differences in pre- and post-test scores were evaluated for statistical significance using the Wilcoxon signed-rank test.

When assessing the participants' scores after the microplanning session, the elements maintained the same rank order as in in the pre-workshop for the mean score for each metric, which were in descending order (post-test mean ± SD): roles and responsibilities (4.61 ± 0.61), perceived engagement (4.45 ± 0.79), SAs (4.39 ± 0.82), and past performance (4.37 ± 0.72).

Fig 2 shows the evolution of pre-post responses for each respondent, among complete pairs. When comparing pre- and post-workshop scores, statistically significant improvements were seen in scores across all four evaluation metrics (p <0.001). The smallest improvement was seen among respondents' understanding of their roles and responsibilities with 0.71 ± 0.95, while the greatest improvement was seen in understanding of past performance with a mean improvement of 1.34 ± 1.05 (Table 3).

The following alluvial charts show the individual-level dynamics in rating before and after the stakeholder workshop. On the left of each chart, the distribution of rating before the workshop with 1 representing "poor" and 5 representing "excellent" is displayed, while the numbers on the right of the chart indicate the responses after the workshop. The thickness of the band represents the number of people giving a particular response combination. The color of the band indicates the response at the time point before the workshop; with orange/red colors representing lower before workshop responses.

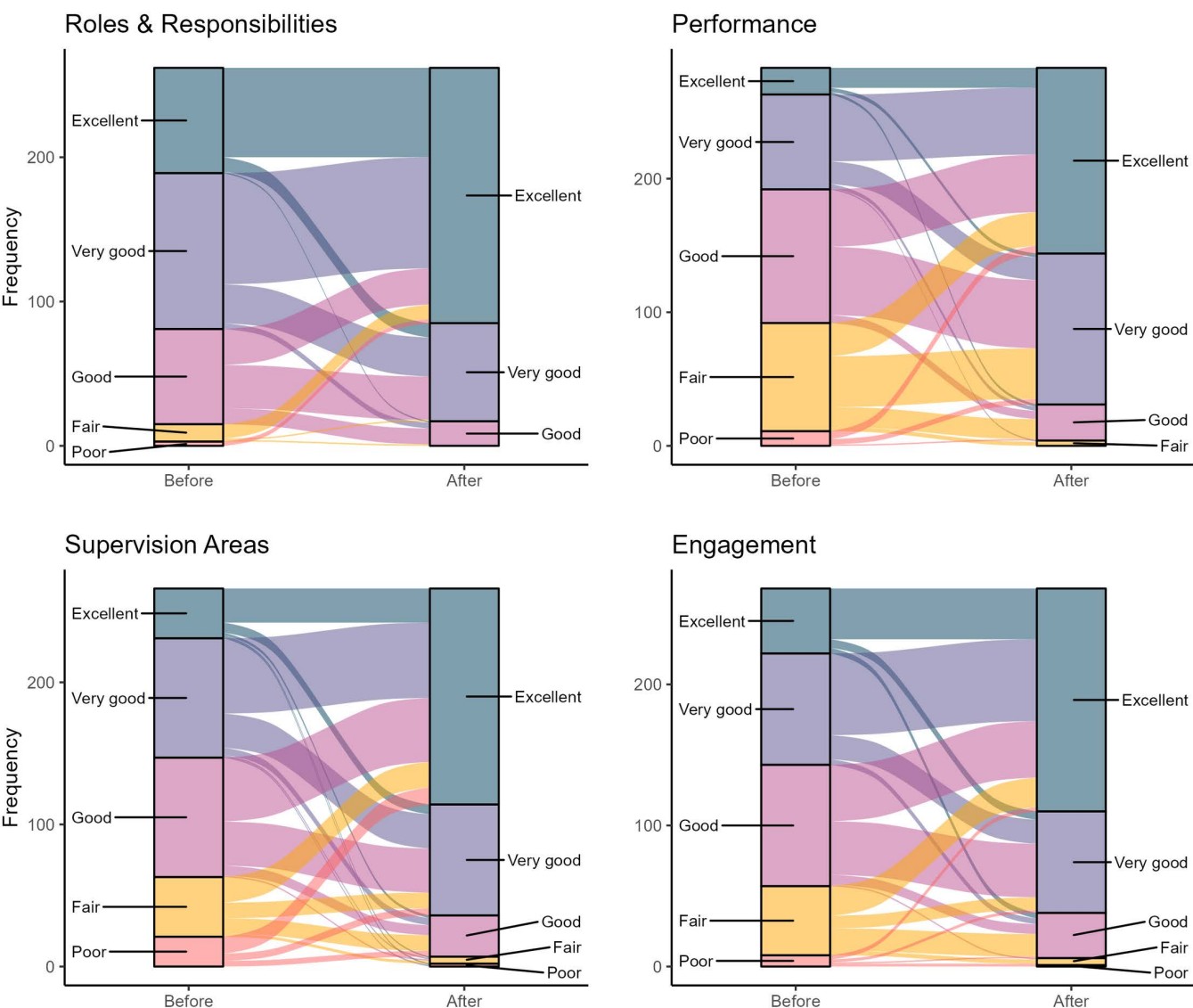

**Fig 2. Changes in participant understanding and perceptions before and after microplanning workshops, Port-au-Prince, Haiti in 2018.** The alluvial charts track the individual-level dynamics in ratings before and after the microplanning workshops for four indicators: (a) understanding respective roles and responsibilities in the MDA; (b) understanding of past performance; (c) understanding of SAs/zones of influence; and (d) perception of engagement by MSPP in the MDA planning process. The left of each chart displays the distribution of participants' ratings before the workshop, while the right indicates the participant responses from after the workshop. Participants ranked their perception on a 5-point scale where: 1 = poor; 2 = fair; 3 = good; 4 = very good; and 5 = excellent. The width of the band corresponds to the relative weight of persons with a particular before-after combination. The color of the band corresponds to the score at the before timepoint, with red/orange colors representing lower before workshop responses.

## MDA coverage

Based on results of the microplanning meeting, the 2018 MDA campaigns in all five communes were extended from 4 to 5 days. MDAs were held from April 26 to June 4, 2018, sequentially in Tabarre, Cité-Soleil, Port-au-Prince, Carrefour, and Delmas. Reported MDA coverage increased in all five communes between 2017 and 2018, respectively (Table 4).

**Table 4. Comparison of mass drug administration (MDA) results prior to and following microplanning workshops held in 2018, Port-au-Prince, Haiti, 2017-2018.**

| Commune | 2017 MDA | | | 2018 MDA | | |
|---|---|---|---|---|---|---|
| | Number treated | Target population | Coverage | Number treated | Target population[1] | Coverage |
| **Carrefour** | 291,007 | 526,948 | 55% | 382,898 | 534,762 | 72% |
| **Cité-Soleil** | 200,659 | 273,160 | 73% | 257,428 | 277,211 | 93% |
| **Delmas** | 268,081 | 537,382 | 50% | 379,691 | 545,351 | 70% |
| **Port-au-Prince** | 552,067 | 1,017,436 | 54% | 835,305 | 1,032,524 | 72% |
| **Tabarre** | 99,335 | 134,258 | 74% | 162,859 | 136,249 | 120% |

[1]Target population was calculated by applying a 1.5% growth rate to the 2015 IHSI population estimates figures for each commune. Source: Unité d'Etudes et de Programmation (UEP), Ministère de la Santé Publique et de la Population (MSPP); Estimation Population Haïti 2015-2020.

## Discussion

### Microplanning outcomes

These evaluation results suggest that microplanning led to several important outcomes. First, the lack of clearly defined and communicated SA boundaries was a key challenge to ensuring broad access to the MDA in previous years. Using microplanning, the microplanning teams were able to collaboratively define SA boundaries with key stakeholders, thus minimizing the likelihood of missing underserved areas, efficiently allocating scarce MDA resources, and avoiding duplication of efforts of a limited workforce across the Port-au-Prince region. With this knowledge, stakeholders were able to collaboratively realign SA boundaries and commit to deploying posts within their respective SAs. As has been seen elsewhere [27,47], the use of digital mapping tools was critical for helping MDA stakeholders to visualize the targeted MDA area and harmonize boundaries of their SA, without which harmonizing SA borders would have been extremely challenging in this densely populated urban setting. Zonal maps provided needed documentation that allowed CLs and CPs to plan and execute activities in their assigned areas, as well as more effectively coordinate with volunteers in neighboring areas. The presence of numerous landmarks in urban settings and orienting microplanning participants to their individual zone helped with the uptake and use of the maps by CLs and CPs. Similar to our findings in Haiti, the process of defining and validating operational boundaries through stakeholder engagement has been shown to be critical for the success of routine immunization services, by allowing efficient allocation of resources, identifying hard-to-reach populations, and reaching people who had not previously been vaccinated [20,27,36,48].

Second, the participatory workshops provided a forum for engaging various stakeholders in identifying key MDA challenges and collectively designing locally appropriate, feasible, and acceptable solutions. Improving stakeholder engagement and using local knowledge to generate appropriate and acceptable solutions for health programming, has been seen in many health program contexts where microplanning has been used [19,28,33,49,50]. For example, MDA volunteers from Port-au-Prince commune revealed that they did not cover certain areas, mainly economically and underdeveloped segments within the commune, due to safety concerns. In light of this, a CL from Cité-Soleil volunteered to cover these areas to ensure that no service gaps existed. Furthermore, volunteers from the Delmas commune expressed frustrations of being scheduled as the final commune to undergo MDA in the area as well as concern that CLs from neighboring communes placed posts within their commune boundaries.

They feared that these factors artificially reduced their coverage figures, as members of the population were already treated by neighboring communes. These concerns were supported by coverage data showing progressively decreasing coverage figures as the MDA progressed sequentially across communes. The refinement of SA boundaries and creation of zonal maps reduced the risk of cross-commune post deployment.

Third, the microplanning activities provided previously unknown data and information that were helpful in supporting other MDA strengthening initiatives. For instance, the distribution post census allowed for the generation of unique identification codes for each distribution post. These codes were then used to track post-based distribution data through real-time data collection, which permitted MDA leadership to track the progress of MDA in near real-time and adapt MDA procedures to achieve adequate coverage. Results of other microplanning exercises have shown that GPS coordinates generated during microplanning were vital for performing real-time monitoring of health team performance [24,27,36,48,51], and that while geographic information system (GIS)-supported microplanning involved additional resources, technology-supported microplanning has been shown to improve program outcomes [22,47] and may be more cost-effective compared with traditional microplanning approaches [24,35].

Finally, one of the most successful outcomes of the microplanning was an increase in community engagement, program ownership, and motivation of MDA volunteers and stakeholders, which has been see as a key output of microplanning in other contexts [19,20,28,49,50]. Though the process of working with CLs and other stakeholders to review and validate SA boundaries, stakeholders felt an increased sense of ownership by being included in highly technical work, rather than serving as passive actors. For example, several leaders took an initiative to conduct informal, uncompensated mop-up activities beyond their allocated MDA days. Increased motivation was corroborated by the results of the pre- and post-workshop surveys from participants attending the microplanning workshop. The survey data provided additional insights into the perceived utility of such participatory planning efforts from the perspective of MDA stakeholders. For example, although LF program leadership perceived themselves to be generally successful in conveying the MDA roles and responsibilities to the various MDA actors, they reported further improvements after the microplanning exercise when roles and responsibilities were reviewed and validated collectively. This finding is supported by microplanning experiences from immunization activities showing that defining roles and responsibilities is a complex yet critical task, particularly in urban contexts [52]. Due to the various actors who may be implicated in urban areas (e.g., municipal, district, and central), failure to define these roles and clarity of who is responsible for each action leads to confusion and ultimately results in certain populations not being adequately served.

The workshop evaluation supported research by Wodnik et al. with CLs, CPs, and CDDs in Tabarre and Carrefour communes [40]. Their evaluation suggested that the LF program needed to improve communication and feedback of the outcomes of the MDA rounds with the stakeholders, as well as provide opportunities for collectively identifying and implementing strategies to improve MDA [40]. Further, their evaluation revealed that SA boundaries were poorly understood by CLs, likely leading to missed populations during previous MDA activities. Finally, they suggested that strategies such as microplanning can be a successful approach for engaging MDA staff and can contribute to increased program ownership and improved MDA outcomes.

We believe that the outputs from this microplanning exercise provides important lessons for other complex urban areas, in Haiti and beyond, aiming to increase MDA coverage. The strategy employed in this project was tailored to Port-au-Prince, based on known challenges in the area, and considered available local resources and needs. We trust the ability to follow

up the microplanning exercises with modified program strategies, such as developing new social mobilization tools or distribution methods, in response to the results of the exercise was essential. However, this requires that sufficient time is available between the microplanning exercise and the MDA in order to implement these changes.

The exercise also highlighted the importance of programs challenging their assumptions about the reasons for low MDA coverage. The microplanning strategy uncovered barriers to MDA compliance that were unexpected and previously unknown challenges. This allowed stakeholders to leverage local knowledge to identify solutions that were acceptable to MDA volunteers and targeted communities. This is consistent with findings from other studies that have suggested that such grassroots engagement can yield a deeper understanding of the MDA, lead to a shared management of resources and responsibilities, and can ultimately lead to higher treatment coverage [13,51,53].

This evaluation had some limitations. Since Port-au-Prince is a large and complex urban setting, it was not possible to engage all relevant actors in the microplanning workshops (e.g., private medical sector who are influential on many individuals' health decisions making or management of gated communities who determine the ability to access these communities). The microplanning evaluation surveys were limited by self-reporting of performance and engagement indicators, which may be subject to social desirability bias, and hence, we cannot assert that microplanning led to increases in specific knowledge or performance metrics as microplanning was included as a broader MDA improvement initiative. Thus, it is not possible to attribute improvements in 2018 MDA coverage to any one improvement strategy. However, the increase in MDA coverage in all five communes in 2018 after years of gradual decline and the use of microplanning results in other MDA improvement strategies supports our belief that microplanning played a key role in improving MDA drug coverage in Port-au-Prince in 2018. Coverage was highest in the earliest treating communes and decreased sequentially, which we hypothesize was due to individuals from neighboring communes being treated and highlights the challenges of assessing coverage during post-based MDA in complex urban settings. Finally, based on limited experience using microplanning in the context of NTDs, it is difficult to comprehensively compare these findings to other contexts.

We believe the results of this evaluation complements other assessments showing that microplanning is a feasible [24,50,54] and effective [22–24] strategy for public health programs that should be considered in areas that are undergoing MDA for NTDs, particularly areas that have had historic challenges in achieving adequate MDA coverage. Microplanning may be particularly useful in settings where it is challenging to define the boundaries of SAs, and where the program has faced challenges in providing feedback to MDA volunteers and staff.

## Supporting information

**S1 Fig.  Analysis of access of the population to distribution posts during the 2017 mass drug administration (MDA) in Delmas commune in Port-au-Prince, Haiti.** An illustrative map generated by the microplanning team to estimate the accessibility of distribution posts for the Delmas population. Global position system (GPS) coordinates of the 2017 MDA distribution posts, represented as black dots, were mapped for the commune of Delmas. The microplanning team aimed for participants to walk no further than 500 meters to reach a distribution post, a measure of distribution post access. A 300-meter buffer (green) was placed around each distribution post, which corresponded to an approximate 500 meter walking distance (Manhattan distance). Areas within Delmas that were not covered by the post buffers were targeted for re-allocation during the 2018 MDA. This map was created using QGIS. The base layer is available

for download at: https://www.openstreetmap.org/#map=9/18.808/-72.905, under the following terms and conditions: https://osmfoundation.org/wiki/Terms_of_Use
(TIF)

**S2 Fig. Final supervision area (SA) boundaries for 2018 mass drug administration (MDA) for community leaders (CLs) by commune in Port-au-Prince, Haiti.** A map generated by the microplanning team in consultation with community leaders during microplanning workshops detailing SAs for each CL for five communes in Port-au-Prince. Areas are color coded by commune with shades of blue representing Carrefour, red representing Port-au-Prince, yellow representing Delmas, purple representing Tabarre, and green representing Cité-Soleil. This map was created using QGIS. The base layer is available for download at: https://www.openstreetmap.org/#map=9/18.808/-72.905, under the following terms and conditions: https://osmfoundation.org/wiki/Terms_of_Use
(TIF)

**S1 Dataset. Deidentified dataset that was utilized in the analysis.**
(XLSX)

## Acknowledgments

We would like to give special thanks to the community drug distributors, community promoters, community leaders, and other stakeholders who enthusiastically participated in the microplanning sessions and for their everyday efforts to eliminate lymphatic filariasis in Haiti. We would also like to thank our partners, namely the Ministry of Public Health and Population (MSPP) staff, RTI International, the Carter Center, U.S. Agency for International Development (USAID), the Pan-American Health Organization (PAHO), and the University of Notre Dame for their collaborative efforts to improve MDA coverage in Port-au-Prince. We would like to thank Dr. Ryan E. Wiegand and Dr. Andrew Hill for their assistance in generating the alluvial diagrams. Further, we would like to thank Andrew Berens from CDC's Geospatial Research, Analysis, and Services Program (GRASP) for creating and printing the supervision area maps for community leaders that were used as part of this project.

## Author contributions

**Conceptualization:** Caitlin M. Worrell, Tara A. Brant, Alain Javel, Eurica Denis.

**Data curation:** Caitlin M. Worrell, Alain Javel, Eurica Denis.

**Formal analysis:** Caitlin M. Worrell, Tara A. Brant.

**Funding acquisition:** Caitlin M. Worrell, Tara A. Brant.

**Investigation:** Caitlin M. Worrell, Tara A. Brant, Alain Javel, Eurica Denis, Carl Fayette, Franck Monestime.

**Methodology:** Caitlin M. Worrell, Tara A. Brant.

**Software:** Alain Javel.

**Supervision:** Ellen Knowles, Cudjoe Bennett, Jean-Frantz Lemoine.

**Visualization:** Caitlin M. Worrell, Tara A. Brant.

**Writing – original draft:** Caitlin M. Worrell, Tara A. Brant.

**Writing – review & editing:** Caitlin M. Worrell, Tara A. Brant, Alain Javel, Eurica Denis, Carl Fayette, Franck Monestime, Ellen Knowles, Cudjoe Bennett, Jürg Utzinger, Peter Odermatt, Jean-Frantz Lemoine.

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
