## [Decision Letter · Decision Letter 0]

31 May 2024

Dear Ms. Worrell,

Thank you very much for submitting your manuscript "Microplanning improves stakeholders’ perceived capacity and engagement to implement lymphatic filariasis mass drug administration" for consideration at PLOS Neglected Tropical Diseases. As with all papers reviewed by the journal, your manuscript was reviewed by members of the editorial board and by several independent reviewers. The reviewers appreciated the attention to an important topic. Based on the reviews, we are likely to accept this manuscript for publication, providing that you modify the manuscript according to the review recommendations. 

Sincerely,

Uwem Friday Ekpo, PhD

Academic Editor

Eva Clark

Section Editor

Reviewer's Responses to Questions

**Key Review Criteria Required for Acceptance?**

**Methods**

-Are the objectives of the study clearly articulated with a clear testable hypothesis stated?

-Is the study design appropriate to address the stated objectives?

-Is the population clearly described and appropriate for the hypothesis being tested?

-Is the sample size sufficient to ensure adequate power to address the hypothesis being tested?

-Were correct statistical analysis used to support conclusions?

-Are there concerns about ethical or regulatory requirements being met?

Reviewer #1: see attached comments

Reviewer #2: Method

The objectives of the study clearly stated and include; to identify and address gaps in geographic coverage of supervision areas by reviewing the past MDA performance and proposes strategies to improve access to MDA.

The design of the study is appropriate. The sample size of the participants, stake holders and numbers of stakeholders meeting in the past MDA performance should be stated for a good comparison with the result of 2018 MDA performance. Lymphatic filariasis required an effective round of MDA with epidermiological coverage of at least 65% for a minimum of 5 rounds (5 years). The epidermiological coverage is determine by 

the number of individual ingesting the medicine at implementation unit(iu) 

population of people living in iu (eligible population targeted for MDA)

to determine the gap in MDA coverage, there is need to state the targeted population of people living in the implementation unit and the number of individual ingesting the drug.

The statistical analysis method is appropriate but the post and pre-workshop differences should be tested with T-test at 95% confident level. There is need for ethical clearance both in the community and school base stakeholders atleast informed consent and permission should be sort from appropriate authorities.

**Results**

-Does the analysis presented match the analysis plan?

-Are the results clearly and completely presented?

-Are the figures (Tables, Images) of sufficient quality for clarity?

Reviewer #1: see attached comments

Reviewer #2: Result

The tables and figures seems to be inadequate, there should be table indicating the previous gap and table addressing the gap in the 2018 microplanning improvement of stakeholders capacity and implementation. The figures and tables presented are clear but not sufficient.

**Conclusions**

-Are the conclusions supported by the data presented?

-Are the limitations of analysis clearly described?

-Do the authors discuss how these data can be helpful to advance our understanding of the topic under study?

-Is public health relevance addressed?

Reviewer #1: see attached comments

Reviewer #2: Conclusion

The discussion are not strongly supported by other people works, all the same it is understandable that not much work has been done in the areas of microplanning. However, the limitation of the analysis should be clearly stated.

**Editorial and Data Presentation Modifications?**

Reviewer #1: see attached comments

Reviewer #2: Editorial and Data Presentation Modifications

Data on the previous result of MDA coverage before microplanning that improve stakeholders capacity and engagement, should be compared with the data of MDA coverage after improved microplanning. If this modification is done, the paper should be accepted

**Summary and General Comments**

Reviewer #1: see attached comments

Reviewer #2: Summary 

The title is adequate, there is need to identify the gaps in geographic coverage and MDA coverage and compare the previous with the result of 2018. A total of 356 stakeholders across 5 communities participated in the 2018 workshop, how many were involved in the previous work. The discussion should be strengthened. There is need to review the article.

General Comments

Line 56- Insert "population" before entire

Line 59 delete "if they desire"

Line 76- Triple drug regimen : Mention the names of the medicine,

Line 171 - Metion target population in the community and school, Examples, No. Of school

Line 224 - 80.34 Gps were collected....Specify the number of participants,

Line 229 - 356 participants; Not same in day 1, Why?? 

Line 230- Specify target population,

Line 304- Authors can go further to test if the mean difference observed between pre and post workshop were significance at 95% confidence level, subjecting their mean/sd values to test.

Line 383-348; Recast, 

Line 401-403 - Recast.

Discussion from line 348-392- should be supported with citations and discuss properly References - Recheck references comply with the journal guideline especially with the use of " _et al_" and in formatting the reference list.

Recommendation: The paper should be considered for publication after the corrections.

PLOS authors have the option to publish the peer review history of their article (what does this mean? ). If published, this will include your full peer review and any attached files.

**Do you want your identity to be public for this peer review?** For information about this choice, including consent withdrawal, please see our Privacy Policy .

Reviewer #1: Yes: Doris W Njomo

Reviewer #2: Yes: Professor Lawrence Patrick Usip

Figure Files:

Data Requirements:

Reproducibility:

References

---

## [Decision Letter · Decision Letter 1]

28 Nov 2024

PNTD-D-24-00455R1Microplanning improves stakeholders’ perceived capacity and engagement to implement lymphatic filariasis mass drug administrationPLOS Neglected Tropical Diseases Dear Dr. Worrell, Thank you for submitting your manuscript to PLOS Neglected Tropical Diseases. After careful consideration, we feel that it has merit but does not fully meet PLOS Neglected Tropical Diseases's publication criteria as it currently stands. Therefore, we invite you to submit a revised version of the manuscript that addresses the points raised during the review process. Please submit your revised manuscript within 30 days Dec 28 2024 11:59PM. If you will need more time than this to complete your revisions, please reply to this message or contact the journal office at plosntds@plos.org. Please include the following items when submitting your revised manuscript:* A rebuttal letter that responds to each point raised by the editor and reviewer(s). You should upload this letter as a separate file labeled 'Response to Reviewers '. This file does not need to include responses to any formatting updates and technical items listed in the 'Journal Requirements' section below.* A marked-up copy of your manuscript that highlights changes made to the original version. You should upload this as a separate file labeled 'Revised Manuscript with Track Changes '.* An unmarked version of your revised paper without tracked changes. You should upload this as a separate file labeled 'Manuscript '. If you would like to make changes to your financial disclosure, competing interests statement, or data availability statement, please make these updates within the submission form at the time of resubmission. Guidelines for resubmitting your figure files are available below the reviewer comments at the end of this letter. We look forward to receiving your revised manuscript. Kind regards, Eva ClarkSection EditorPLOS Neglected Tropical Diseases

Shaden Kamhawi

co-Editor-in-Chief

Paul Brindley

co-Editor-in-Chief

 **Journal Requirements:**

Please amend your detailed Financial Disclosure statement. This is published with the article. It must therefore be completed in full sentences and contain the exact wording you wish to be published. Please ensure that the funders and grant numbers match between the Financial Disclosure field and the Funding Information tab in your submission form. Note that the funders must be provided in the same order in both places as well. State what role the funders took in the study. If the funders had no role in your study, please state: "The funders had no role in study design, data collection and analysis, decision to publish, or preparation of the manuscript.".

  **Reviewers' comments:** Reviewer's Responses to Questions

**Key Review Criteria Required for Acceptance?**

**Methods**

-Are the objectives of the study clearly articulated with a clear testable hypothesis stated?

-Is the study design appropriate to address the stated objectives?

-Is the population clearly described and appropriate for the hypothesis being tested?

-Is the sample size sufficient to ensure adequate power to address the hypothesis being tested?

-Were correct statistical analysis used to support conclusions?

-Are there concerns about ethical or regulatory requirements being met?

Reviewer #3: Abstract

Line 41: “ranked their attitudes…” Attitudes towards what?

Lines 49-50: “…drug coverage increased…” Increased from 2017 MDA or all previous years of MDA?

Introduction

Lines 78-82 “Triple drug therapy…” Is IDA used in Haiti? If not, may not be a relevant statement. If it is, may wish to state that it is the strategy used in Haiti.

Line 89: this is a really small point, but the phrase “…the majority of community member who did not receive the medication…” the “the” before medication is a little awkward, because readers of the present paper may not know what medication is being referred to if not involved in NTDs. Suggest simply removing the “the” before medication.

Lines 89-91: In my reading of the paper cited, the most common reason for not being treated was absence at the time of the distribution. If this was related to “inopportune timing”, would suggest “…inopportune timing of the MDA which meant that many people were at their farms at the time of the MDA” (if I’m correctly recalling that paper).

Line 92: “insufficient time…” Insufficient time for CDDs to reach their distribution targets or cover their distribution areas?

Lines 110-113: “the create a realistic operational plan…as well as determine the material, financial, and human resources…” Most NTD programs conduct some sort of microplanning, though typically not in the way described in this paper. Instead, it’s typically a district compiling information provided by health center nurses, which are then rolled up to a district’s needs and then sometimes to regional needs. I’ve heard pushback from Program when discussing a more community-led or -involved approach that the budgets and resources they would ask for would be unrealistic. I wonder if the authors encountered this during this exercise at all, and if so, how it was managed?

Lines 119-120: “Neglected tropical disease programs…have applied microplanning less frequently…” I’m not certain NTD programs (or their partners) would entirely agree with this sentence as it reads now, as all of those I’ve worked with do some sort of microplanning prior to MDA. However, I would agree with this statement if referring to the community-level aspect/mapping aspects, as this often has not been part of developing these plans. Relatedly, is there a reference supporting the statement about financial constraints?

Line 123: Given that the WHO NTD microplanning manual is meant to be disease agnostic, suggest removing “LF” and replace with “disease” or insert some language, such as “…or in the example of LF, assessments show that transmission remains despite multiple rounds of treatment.”

Lines 125-128: “Increasingly, digital tools…” Using digital tools really seems to be a central part of the microplanning process in Haiti and maybe why it was successful, but when I first read this section of the introduction, I wondered if these sentences even belonged, because they seemed to be tossed in for good measure. I wonder if pulling this into a separate paragraph and maybe talk specifically about the need to employ these tools in Haiti would be helpful and the help the reader to orient themselves to the fact that this was actual central to the microplanning process in the way Haiti conducted it.

Line 132: “initial rounds” can this be defined?

Line 138 “…the partners identified…” would it be better to state “the MSPP and its partners…”? otherwise, the language makes it sound as though the MSPP had no involvement in identifying strategies. In this case, the authors may be referring to “partners” as both the MSPP and implementing partners, but given that the term “implementing partners” was used in the previous sentence separately from the MSPP, it is not clear who was involved.

Methods

Line 153: the font size used for the CDC human protection project identification number is different to the font used elsewhere.

Line 162: “…is allocated to treat approximately 1,000 people…” allocated enough drug? Is asked to treat, 1,000 people?

Lines 173-176: “we applied a 300 m buffer…” This sentence is not clear to me. Could this be shown in a figure?

Line 184: “voodoo temple” the other representatives listed are job titles/leadership roles, but “voodoo temple” is a place. Could it be “voodoo temple leaders” or “voodoo temple priests”? I’m not sure what the right title would be, but an actual position, job, or title seems to be needed here.

Lines 186-191: in these few lines, the terms “MSPP staff,” “implementing partners,” “project staff,” and “program staff” are all used. IP and MSPP staff are clear—but not clear what is meant by “project staff” and “program staff.” May need to look at wording throughout so it’s clear what was the role of the MSPP and the role of the IP (and, if there are more categories, their roles).

Lines 199-204: the text indicates that retrospective pre-testing reduces response shift bias. I have to admit, this is new terminology to me, but is there not a possibility that a social desirability bias could have been introduced, in which participants rate their new attitudes/understanding higher and their old understanding lower, so that the Program and Partners feel that the participants learned more than maybe they actually did?

Lines 206-207: “engagement by MSPP in the MDA planning process.” It’s not clear what this measure means or exactly why this measure was used, and I’m wondering: is it a fair question to ask community-level workshop participants? Would they either really feel comfortable indicating low MSPP engagement or would they have a perception of what “MSPP engagement in the MDA planning process” entails?

Lines 263-264: “…including some modifications proposed by LF program staff…” This part of the sentence reads awkwardly. First, were LF program staff not instrumental in all modifications? Second, it introduces yet another term as per comment above re: “MSPP staff,” “project staff,” “program staff.”

**Results**

-Does the analysis presented match the analysis plan?

-Are the results clearly and completely presented?

-Are the figures (Tables, Images) of sufficient quality for clarity?

Reviewer #3: Figure 1

I appreciate that this figure was modified to ensure privacy, and description of the map indicates that it’s been lightly edited to anonymize the names of the CLs, but it seems the whole map is blurry. If it’s meant to be blurry to maintain anonymity, suggest making that explicit in the description.

Figure 2

There doesn’t appear to be any discussion about respondents who ranked their knowledge/understanding of the 4 areas evaluated as lower in the second part of the retrospective pre-test (i.e. after the intervention). The numbers are fairly small, but it is curious that, for example, there were persons who stated their knowledge of supervisory areas was excellent prior to the microplanning process but poor afterwards. Could the authors include this in the discussion?

Table 1

2nd column, 4th row: “working conditions should be improved…” the decision/actions don’t really seem to be about improved working conditions but about having the CL and CP positions made official and therefore recognized by communities. “working conditions” to me sounds like labor law issues.

2nd column, 10th row: “limited school information is available to staff…” which staff? MSPP staff? LF program staff? Other? Second, is the issue here that amounts of drug needed for schools can’t easily be quantified because the school populations are not known? Might want to make that explicit.

2nd and 3rd columns column, 15th row “several areas had insufficient distribution post coverage” It’s not clear by the solution exact issue was—is it that more distribution posts are needed because of the mountainous areas? Is it that the distribution posts don’t maximize where people actually live?

2nd and 3rd columns, 16th row: should “maximum” be “maximize”? and, will mop-up be implemented everywhere? If not was there a way suggested to prioritize mop-up areas?

Line 300: “several challenges…were unable to be addressed fully…” suggest brief reason for this is included and/or whether these issues will be reviewed at a later date.

Line 311: there appears to be an errant comma between “the” and “microplanning”. If you feel a comma is needed, suggest it is moved after “following.”

Table 4: The increase in coverage in Tabarre was quite remarkable and might warrant some further discussion. This was somewhat explained by Tabarre conducting its MDA first in the 2018 MDA—but was this the only reason? And, given that the reported coverage is well over 100%, does this either indicate denominator issues (and why would this only be a problem in Tabarre), or does it potentially indicate ongoing issues of supervisory areas?

**Conclusions**

-Are the conclusions supported by the data presented?

-Are the limitations of analysis clearly described?

-Do the authors discuss how these data can be helpful to advance our understanding of the topic under study?

-Is public health relevance addressed?

Reviewer #3: Lines 402-403: “one such area…” is this sentence needed?

Lines 417-422: I think the authors are likely correct in their assessment of the importance of having the GIS to track coverage and reach targets. My question on cost-effectiveness, though, is: would the calculations be the same for NTDs as for vaccines? For example, in citation 35, part of the calculation for cost-effectiveness is in DALYs averted—childhood vaccines avert death or lifelong disability, in many cases, so the DALYs averted would be much higher than for morbidity that typically develops in adulthood. Additionally, since this paper is not showing the economics of microplanning for LF MDA, I might suggest tempering the argument about cost-effectiveness here.

Line 454: “…other areas looking to increase MDA coverage…” Do the authors mean other areas in Haiti, or other settings/other countries/other National Programs?

Lines 458-460: “…this requires that sufficient time is available…” Can the authors outline the timeframe used in Haiti between the microplanning and the MDA to give other programs and partners an idea of how long this process took in Haiti?

**Editorial and Data Presentation Modifications?**

Reviewer #3: (No Response)

**Summary and General Comments**

Reviewer #3: This is an interesting programmatic use of microplanning for NTDs. There has been a surge of interest in examining ways to improve coverage for MDAs, and this adds a nice, practical example of what could be implemented. The paper is nicely written and generally clear; the majority of my comments were based on reading (and re-reading of certain paragraphs) that I could not quite understand as written. That being said, there were a few areas that could use a bit more fleshing out: 1) from just reading the introduction, it was not clear that making maps/use of GIS was very central to the microplanning process; it was only once I got to the methods that I realized that. It would be helpful to bolster this part of the introduction; 2) the authors indicate that they used a "retrospective pre-test" approach to minimize response shift bias, but couldn't social desirability bias have been introduced? If the literature shows otherwise, it would be helpful to discuss this; 3) one of the measures used in the pre-post test was not clear "engagement by the MSPP in the MDA planning process." First, it's not clear to me what the definition used was, and second, whether it was really a fair question to ask/fair area for evaluation when most of the respondents were at the community level. This does require further clarification; 4) finally, in the discussion, the authors talk about digital tools being more cost-effective than traditional microplanning. This part of the discussion falls a little flat for me, given that the paper doesn't examine this at all (but expect the reader to feel that it would be more cost effective in this case as well), and in at least one of the citations given, cost-effectiveness seems to rely on use of DALYs--and I'm not certain that this would be equivalent for LF and vaccine-preventable diseases. But overall, an interesting paper and glad the authors have written this up.

PLOS authors have the option to publish the peer review history of their article (what does this mean? ). If published, this will include your full peer review and any attached files.

**Do you want your identity to be public for this peer review?** For information about this choice, including consent withdrawal, please see our Privacy Policy .

Reviewer #3: **Yes: ** Stephanie L Palmer

---

## [Editor Report · Decision Letter 2]

5 Mar 2025

Dear DR WORRELL,

We are pleased to inform you that your manuscript 'Microplanning improves stakeholders’ perceived capacity and engagement to implement lymphatic filariasis mass drug administration' has been provisionally accepted for publication in PLOS Neglected Tropical Diseases.

Best regards,

Angela Monica Ionica, Ph.D.

Academic Editor

Eva Clark

Section Editor

Shaden Kamhawi

co-Editor-in-Chief

Paul Brindley

co-Editor-in-Chief

---

## [Editor Report · Acceptance letter]

Dear Ms. Worrell,

We are delighted to inform you that your manuscript, "Microplanning improves stakeholders’ perceived capacity and engagement to implement lymphatic filariasis mass drug administration," has been formally accepted for publication in PLOS Neglected Tropical Diseases.

Best regards,

Shaden Kamhawi

co-Editor-in-Chief

Paul Brindley

co-Editor-in-Chief
